# Crankshaft High-Cycle Bending Fatigue Experiment Design Method Based on Unscented Kalman Filtering and the Theory of Crack Propagation

**DOI:** 10.3390/ma16227186

**Published:** 2023-11-16

**Authors:** Tianyi Que, Dongdong Jiang, Songsong Sun, Xiaolin Gong

**Affiliations:** 1College of Automobile and Traffic Engineering, Nanjing Forestry University, Nanjing 210037, China; youben2023@163.com (T.Q.); sunsong1987@126.com (S.S.); 2Li Auto Vehicle Control Operation System, Hangzhou 310000, China; jiangdongdong@zju.edu.cn

**Keywords:** crankshaft, crack propagation, fatigue limit load, unscented Kalman filtering algorithm

## Abstract

The high-cycle bending fatigue experiment is one of the most important necessary steps in guiding the crankshaft manufacturing process, especially for high-power engines. In this paper, an accelerated method was proposed to shorten the time period of this experiment. First, the loading period was quickened through the prediction of the residual fatigue life based on the unscented Kalman filtering algorithm approach and the crack growth speed. Then, the accuracy of the predictions was improved obviously based on the modified training section based on the theory of fracture mechanics. Finally, the fatigue limit load analysis result was proposed based on the predicted fatigue life and the modified SAFL (statistical analysis for the fatigue limit) method. The main conclusion proposed from this paper is that compared with the conventional training sections, the modified training sections based on the theory of fracture mechanics can obviously improve the accuracy of the remaining fatigue life prediction results, which makes this approach more suitable for the application. In addition, compared with the system’s inherent natural frequency, the fatigue crack can save the experiment time more effectively and thus is superior to the former factor as the failure criterion parameter.

## 1. Introduction

In recent decades, modern engines have been widely used in various kinds of industry fields. This equipment can conveniently provide enough power for the application of engineering machinery and also results in high enough strength demands for the main parts of itself, especially for the power transmission parts such as the connecting rod and the crankshaft [1,2]. In addition, the lightweight design methods have been extensively used in engine parts, which also leads to the higher strength demands for these parts [3,4].

Focusing on this problem, relevant work has been carried out in recent years. Among these, Calderon-Hernandez analyzed the fatigue failure mechanism of a selected crankshaft and discovered the main three reasons: the inadequate heat treatment, material selection problems, and the inadequate bushing adjustment that induced the heating and superficial deformation and consequently resulted in the fretting fatigue of the component [5]. Hosseini used the acoustic emission entropy method in researching the fatigue crack property of the crankshaft; in this way, the data volume was greatly reduced and it was much more practical and cost-effective for real-time health monitoring [6]. Yanping Wang analyzed a broken 42CrMo crankshaft from the heavy truck and found that that the crack source was located at the junction of the threaded bottom hole column surface and taper surface, and the main reason for the fracture was the stress concentration caused by the lack of an obvious transition fillet within the same area, as well as the metallographic structure transformation of the surface layer of the bottom hole sidewall caused by the high machining temperature [7]. Shuailun Zhu presented the failure analysis and the numeral simulation work of a given crankshaft with the combination of the relevant response surface optimization; in this way, the stress and deformation of the part was reduced obviously [8]. Leitner conducted a strength assessment of electroslag remelted 50CrMo4 steel crankshafts based on the theory of multi-axial fatigue and proposed an improved model which can predict the fatigue property of a given component more accurately than the previous empirical models [9]. Ahmed combined the low-cycle fatigue test results and the 2D-FE evaluation of the J-integral; in this way, the comprehensive evolution of the fracture features can be exhibited [10]. Qin adopted the critical plane method in analyzing the influence of the residual stress field on the fatigue property, and the proposed model could fit the accuracy demands [11]. Fonte analyzed a failed crankshaft from a diesel motor engine and discovered that the main reason for the failure may not be attributed to the part itself but to the misalignment of the main journals and a weakness in design close to the gear in the region where the crack was initiated [12]. Venicius applied multi-axial fatigue criteria to motor crankshafts in thermoelectric power plants to provide guidance for the selection of the material in the production [13]. Bulut proposed a new fatigue safety factor model to analyze the fatigue life of the crankshaft from a single-cylinder diesel engine under variable forces and speeds; in this way, the comprehensive evaluation of the safety of the crankshaft during the whole working period can be achieved [14]. Khameneh extracted the standard specimens from the crankshaft and examined them with a four-point rotary-bending high-cycle fatigue testing machine; the results indicated that the high-cycle fatigue lifetimes were lower than the S-N curve from the FEMFAT data bank and that the standard specimens extracted from the crankshaft could be used to consider the manufacturing effects [15]. Singh conducted the fatigue life analysis of a diesel locomotive crankshaft and proposed a 3D finite element model to research the relationship between the fillet radius and the least life of the crankshaft. Based on this, the optimum structural design of the crankshaft can be proposed [16]. Fonseca analyzed the influence of the manufacturing process on the residual stress, which was caused by deep rolling with the combination of the finite element analysis and the corresponding fatigue tests. The research results could provide the theoretical basis for the optimization design of the process [17,18]. Antunes analyzed the finite element meshes for the optimal modeling of plasticity-induced crack closure and proposed the analytical expression of the most refined region along the crack propagation area. The results showed that there may be an optimum value for the plasticity-induced crack closure [19].

Currently, the most common fatigue damage type of the crankshaft in modern engines is bending fatigue damage. Therefore, specialized bending fatigue property performance evaluation is indispensable for theoretical guidance before the actual application. At present, this goal can be achieved through the professional bending fatigue bench test [20]. As shown in Figure 1, the main components of the devices are the vibration arms (both the initiative and passive types), the electric motor which is used to provide the dynamic exciting torque load, and the sensors for online monitoring. Because of the resonance effect of the system, during the experimental process, the amplitude of the dynamic bending moment applied on the crankshaft is much larger than that of the initiative bending moment generated by the electric motor. Meanwhile, the fatigue crack caused by the alternating load will appear at the fillet of the upper crankpin and result in a reduction in the system stiffness, as well as the inherent frequency. When the reduction amount of the failure criterion parameter reaches the determined value, the crankshaft is defined as broken [21,22].

As shown in Figure 1, this equipment can provide the specified alternating bending moment on the crankshaft to approximately simulate the load condition of the part arranged in the engine, but the time period may be relatively longer because of the high-cycle fatigue requirement. In addition, the test results in this application scenario always show an obvious dispersion property, which also brings a larger sample size and finally results in a longer time period.

In previous studies, some commonly used algorithm methods, such as particle filtering, were applied in predicting the remaining fatigue life of the crankshaft [15,23,24]. In this way, the load time can be saved, as well as the experiment cost. The results showed that these methods can predict the goal parameter accurately in most situations, but the results may contain obvious errors in some cases, even though the optimized sampling ranges were chosen. The relative error between the prediction and the actual experimental data may be more than 15%, which may be attributed to the particle degeneracy property of the algorithm itself. In addition, the failure criterion parameter of the fatigue test also affects the predicting accuracy obviously, as well as the goal achievement degree of the timesaving effect.

In order to solve this problem, more comprehensive work was carried out in this paper. First, the loading period was quickened through the prediction of the residual fatigue life based on the unscented Kalman filtering algorithm approach and the crack growth speed. Then, the accuracy of the predictions was improved obviously based on the modified training section based on the theory of fracture mechanics. Finally, the fatigue limit load analysis result was proposed based on the predicted fatigue life and the modified SAFL (statistical analysis for the fatigue limit) method. The results showed that the combined method proposed in this article can predict the remaining fatigue life more accurately than the former models to provide nearly the same experimental results. In addition, among the two failure decision parameters, the fatigue crack length is more suitable on account of the more effective timesaving results. The conclusions proposed in this paper can provide some theoretical guidance for the crankshaft manufacturing industry.

## 2. Method

### 2.1. The Acceleration Method

As introduced in the above chapter, during the experimental process, the alternating load provided by the test bench is applied to the crankshaft until it breaks. At present, the fatigue damage type of the crankshaft is high-cycle bending fatigue. For metal components, such as crankshafts made of steel, the high-cycle fatigue life range is defined as between 10^5^ and 10^7^. This makes the experimental period long and expensive. In other words, if the crankshaft can be judged as broken before the final failure time node, the time taken, as well as the experiment cost, can be reduced. Up to now, a significant amount of related work has been carried out in various kinds of industry fields. The remaining life of some industry components, such as the batteries and bearings, has been predicted by various kinds of models. However, the type of life reported in these studies is the working life. Similar research on the fatigue life has rarely been reported [25,26].

At present, in the remaining life prediction process, the remaining life is usually defined as the life period between the prediction time node and the final failure time node. Thus, the correct definition of the failure criterion parameter is necessary for the reasonable evaluation of the fatigue property. In addition, the most commonly used materials applied in the crankshaft manufacturing industry are high-strength alloy steel and spheroidal graphite cast iron. For the components made of these kinds of materials, usually, an obvious fatigue crack propagation phenomenon can be found during the fatigue failure process. Therefore, the remaining fatigue life in this paper can be defined as the number of fatigue load cycles between the given time node and the time node at the limit fatigue crack length [27,28].

On the other hand, according to the related fatigue property studies published in the past decades, a symbolic feature of the fatigue failure process is that the damage amount accumulation process shows an obvious nonlinear property. According to previous studies, the unscented Kalman filtering approach is a suitable choice [29,30].

The detailed process of the prediction is shown in Figure 2.

### 2.2. The Fatigue Crack Determination Method

In this paper, the degree of injury of the crankshaft during the fatigue test process can be evaluated with the fatigue crack length. Figure 3 shows the fatigue fracture surface of the crankpin after the fatigue test, from which a clear conclusion that the shape of the crack surface caused by the alternating bending moment is elliptic can be determined. In addition, the location of the fatigue crack source is the fillet of the crankpin. The main structural parameters of the crack surface shape are shown in Table 1.

In a previous study, we applied the combined finite element analysis and vibration approach to determine the crack growth process throughout the whole experimental period. The main principle of this method is that during the fatigue loading process, the crack propagation will result in a reduction in the system stiffness, which is the primary cause of the inherent frequency reduction in the system [27,28]. So, the length and the width of the crack surface can be determined indirectly by checking the vibration property of the system at the specified time node. The detailed information is shown in Figure 4.

On the basis of this method, the main crack surface structural parameters at any time node can be measured indirectly with the vibration property of the system at the same time node; the results are shown in Table 2.

As introduced in the experimental method section, when the damage degree has accumulated to the predetermined value, the crankpin is judged as broken. At present, the standard is when the decrement of the first-order inherent frequency has increased by 1 Hz. Under such circumstances, the crack length of the broken crankshaft can be determined to be 10 mm. Thus, the remaining fatigue life prediction in these cases can be proposed in two steps. First, the parameters of the UKF model are determined by training the data within the specified training range. Then, the remaining fatigue life is predicted based on the known UKF model and the subsequent crack growth process.

### 2.3. The Fatigue Experimental Data Analysis Method

As mentioned in the published materials, one of the basic features of the fatigue experimental result is the obvious dispersion property. On the other hand, compared with that of the object of simple constructions, the experimental cost of the complicated parts is usually much higher. Therefore, it is necessary to propose a corresponding data processing method to analyze the fatigue experimental results with a relatively small sample size [31]. At present, the most commonly used method is the statistical analysis method for the fatigue limit load; the corresponding theoretical basis is shown in Figure 5.

As shown in Figure 5, the fatigue life and the load amplitude in each experimental case are exhibited in the double logarithmic coordinate system. The point A is the fatigue load value obtained through fitting the relationship between them based on the least square method and the specified low fatigue life point. Thus, the distribution property of the fatigue limit load at the limit of fatigue life can be proposed by mapping the fatigue load amplitude from the point A. The expression in each case can be calculated as follows:(1)$$\mathrm{lg}C_{i}=\mathrm{lg}F_{A}\frac{\mathrm{lg}N_{0}-\mathrm{lg}N_{i}}{\mathrm{lg}N_{A}-\mathrm{lg}N_{i}}+\mathrm{lg}F_{i}\frac{\mathrm{lg}N_{0}-\mathrm{lg}N_{A}}{\mathrm{lg}N_{i}-\mathrm{lg}N_{A}}$$
where $F_{A}$ is the load amplitude of the point *A*. In this paper, the predicted fatigue life takes the actual experimental result instead in the given case for the analysis. In this way, the comprehensive evaluation of the applicability of this method can be carried out in detail.

## 3. Result and Discussion

### 3.1. Remaining Fatigue Life Predictions

Figure 6 shows the experimental results of the selected crankshaft. It can be seen that among the ten groups of the experimental results, only three cases last relatively long (the fatigue life is more than 2 × 10^6^) and require to be shortened. The changing process of the fatigue crack length in these three cases is obtained based on the finite element method introduced in the previous chapter and the online monitoring of the system response. The detailed results are shown in Figure 7.

As shown in Figure 7, it can be discovered that in each case, the values of the crack growth speed increased rapidly when the crack length reached 10 mm, which is in accordance with the failure criterion parameter analysis result in the previous chapter. In addition, the gradient of the fatigue life dependence of the fatigue load was changed above one million cycles. This phenomenon can be explained by the theory of fracture mechanics, according to which the whole process can be divided into three stages: the crack initial stage, the crack steady growth stage, and the crack rapid growth stage. As a result of this, the crack growth speed will increase obviously after a certain cycle. In this paper, three kinds of training and prediction section definition methods are proposed for the comprehensive comparison. The detailed definition of them is shown in Table 3.

As shown in Table 3, in each range, the whole fatigue damage process of the crankshaft is made up of two parts, among which the training section is used to train the UKF model proposed in the previous chapter. Then, the main model parameters determined based on the training can be applied in predicting the fatigue crack growth property in the coming prediction section. In other words, the smaller the size of the training section, the bigger the size of the defined prediction section, as well as the more experimental time can be saved. With the help of the different section definitions, the prediction work can be carried out based on the UKF.

As shown in Figure 8, the predictions in all three groups are obviously different from the actual original experimental data, which makes the predictions based on this section definition completely useless. The main reason for this phenomenon may be attributed to the relatively smaller data size applied in the training section. For the predictions in Figure 9 based on the second training section, the accuracy of the third group has been improved obviously. This phenomenon showed that an increase in the training data can effectively improve the accuracy of the prediction. For the other two groups, the values of the accuracy of the predictions are still too low to be applied to further statistical analysis. Figure 10 shows the predictions based on the third training section; still, obvious differences can be discovered between the predictions and the actual experimental results in all the three groups. Generally speaking, these three kinds of training and prediction section combinations cannot fulfill the actual engineering application demand, which makes the corresponding modification and improvement of the model necessary.

### 3.2. The Improved Model and Application

Based on the above research, it is clear that the direct application of the UKF model in the remaining fatigue life prediction is slightly inappropriate because of the relatively poor accuracy degree. Therefore, a corresponding optimization method should be proposed to improve its applicability in this situation. According to our previous study, the modified training section based on the crack growth speed can effectively solve this problem; thus, this method is also selected to be the object of the improved technology in this paper [23,24]. The detailed information on the improved sections is shown in Table 4.

According to the definitions in Table 4, it is clear that the range of the prediction section in each case is just the same as that in the original model. In other words, the percentage of the saved experimental time period in the total process after this modification still remains unchanged. In addition, the start point of these modified training ranges has been changed from 0 to 1 mm. According to the theory of fracture mechanics, during the crack propagation process, the growth speed of the crack is not constant. The whole process can be divided into three stages: the crack initial stage, the crack steady growth stage, and the crack rapid growth stage. Based on this modification, the crack growth process within the initial stage has been cleared away in advance, and the predictions are just based on the crack growth process within the steady stage.

The predictions based on the first newly proposed training section are shown in Figure 11, which clearly states that the predictions have been obviously enhanced in the degree of precision. For group 1, the error is less than 10%, which can already meet the engineering application requirements. For the other two groups, the values of the errors have been ulteriorly reduced to less than 5%. In addition, similar situations can be found among the predictions based on the other two kinds of training sections (Figure 12 and Figure 13). In conclusion, this modification method can effectively improve the prediction accuracy.

In our published paper, the particle filtering (PF) algorithm method has been selected to be combined with this modification method for predicting the remaining fatigue life [23,24]. Table 5 shows the errors of these two methods in this application situation and the same training section (the first training section); it can be seen that the errors generated by the UKF method are much lower than those generated by the PF model. This situation makes the latter approach more superior to the former in such application scenarios. On the other hand, sometimes the failure criterion parameter in the crankshaft fatigue experiment is selected to be the first-order natural frequency. To ensure that the conclusions proposed in this paper are more credible and comprehensive, corresponding work was also carried out in predicting the same objects, and similar conclusions were found [32]. The degrees of the timesaving percentage based on these two kinds of fatigue failure criterion parameters are also shown in Table 5. It means that although these two kinds of failure criterion parameters can result in a close level of prediction accuracy, the percentage of the saved experimental time in the application of the UKF model is much higher than that of the PF approach. On account of this superiority, the approach proposed in this paper is much more suitable for the application.

### 3.3. Fatigue Limit Load Analysis Results

At present, most engineering equipment must be able to work normally during the designed working period [33,34]. In addition, the service life of a crankshaft is limited to a certain number of cycles depending on the demand of the travelling distance. As a result of this, compared with the common fatigue property evaluation parameter (usually the fatigue life under a given load), it is more important to correctly evaluate the high-cycle fatigue load of a crankshaft under a specified fatigue life [35,36]. According to previous studies, the SAFL (Statistical Analysis for Fatigue Limit) approach is considered to be an effective method for analyzing the distribution property of the fatigue limit load [37]. With the combination of the predicted fatigue life and the SAFL method, it is possible to analyze the fatigue limit load of the crankshaft under different survival rates; the results are shown in Figure 14. The curves within this figure clearly show that the fatigue limit load analysis results according to both the predictions and the actual experimental data are almost the same, which makes the timesaving effect completely achieved (the biggest value of the relative error under any survival rate is less than 1%).

## 4. Conclusions

At present, the high-cycle bending fatigue experiment is widely applied in the crankshaft manufacturing industry. In this paper, the speed of the experiment was quickened using the remaining fatigue life prediction approach. The main conclusions are as follows:

(1)The combination of the UKF model and the training section proposed in this paper can provide a desirable outcome in the remaining fatigue life prediction application, which can be explained by the time varying fatigue damage accumulation rate during this process. In addition, the improved training section can provide much higher accuracy in this condition without affecting the timesaving effect, which makes this modification absolutely valuable.(2)The UKF model can provide much higher precision than the PF method in this application situation. In addition, the more suitable fatigue failure criterion parameter in the same situation is the fatigue crack length because of the higher timesaving percentage. These two factors make the approach proposed in this paper is more superior to the former methods.

## Figures and Tables

**Figure 1 materials-16-07186-f001:**
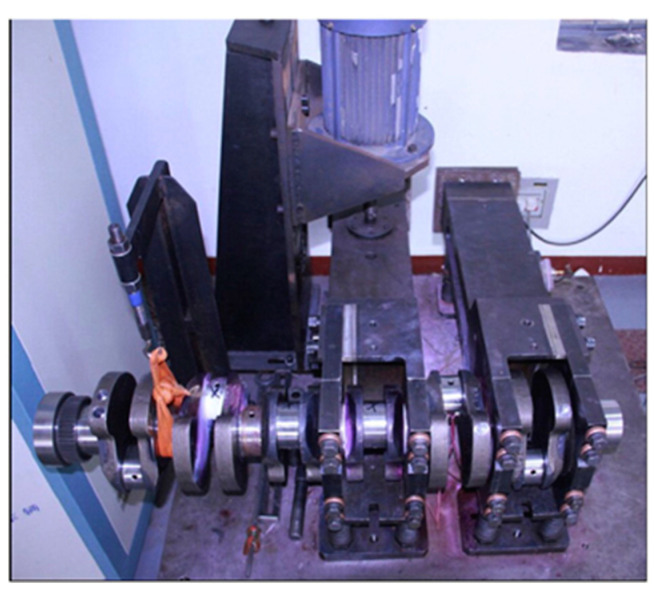
The experiment bench for the crankshaft bending fatigue test.

**Figure 2 materials-16-07186-f002:**
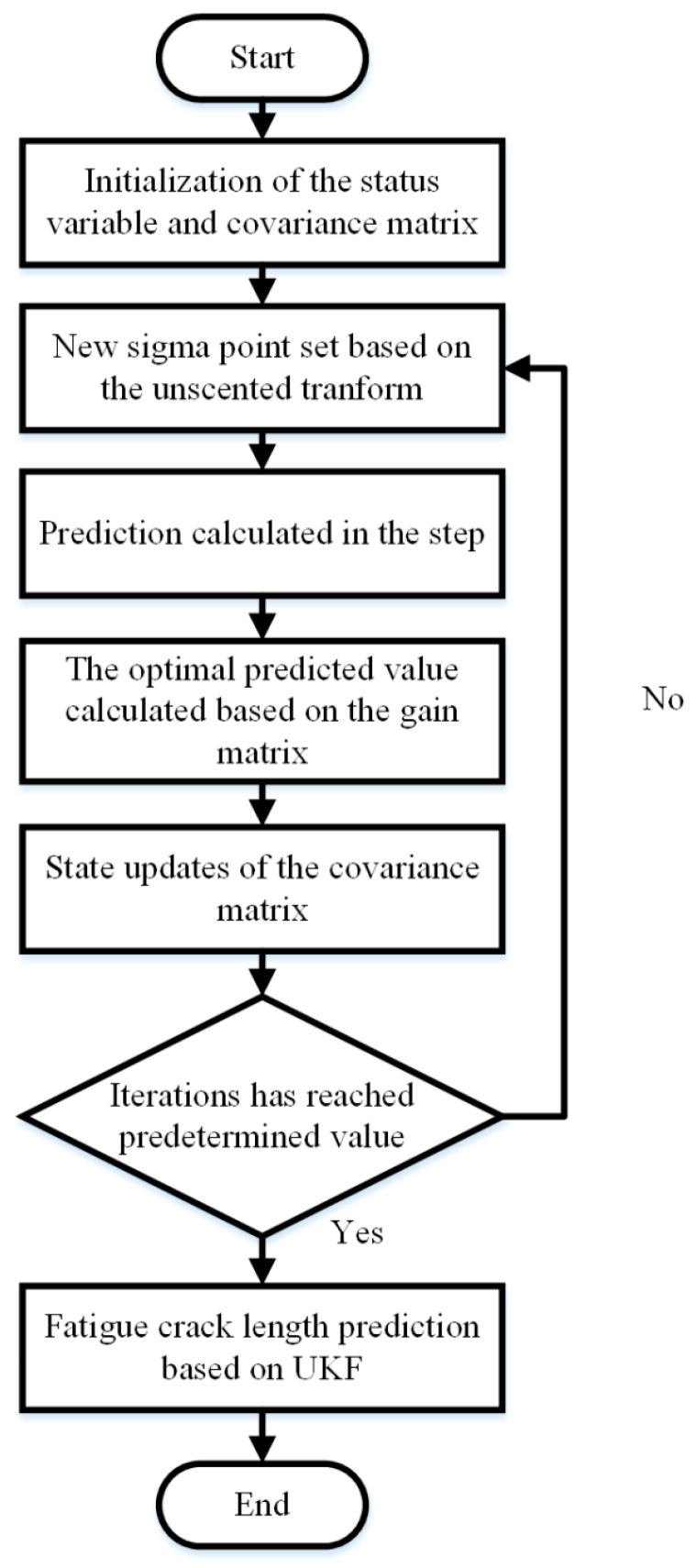
The research process of the UKF method.

**Figure 3 materials-16-07186-f003:**
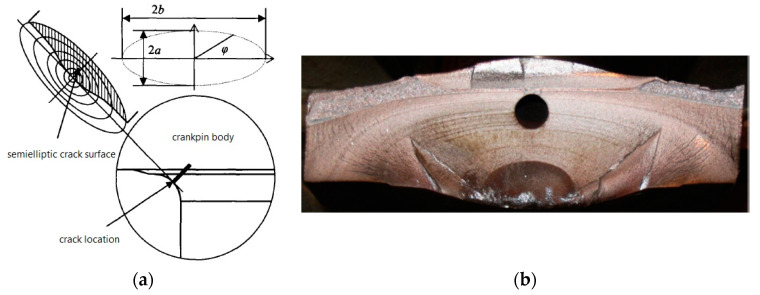
The structural features of the fracture surface ((**a**) the crack shape; (**b**) the fracture surface).

**Figure 4 materials-16-07186-f004:**
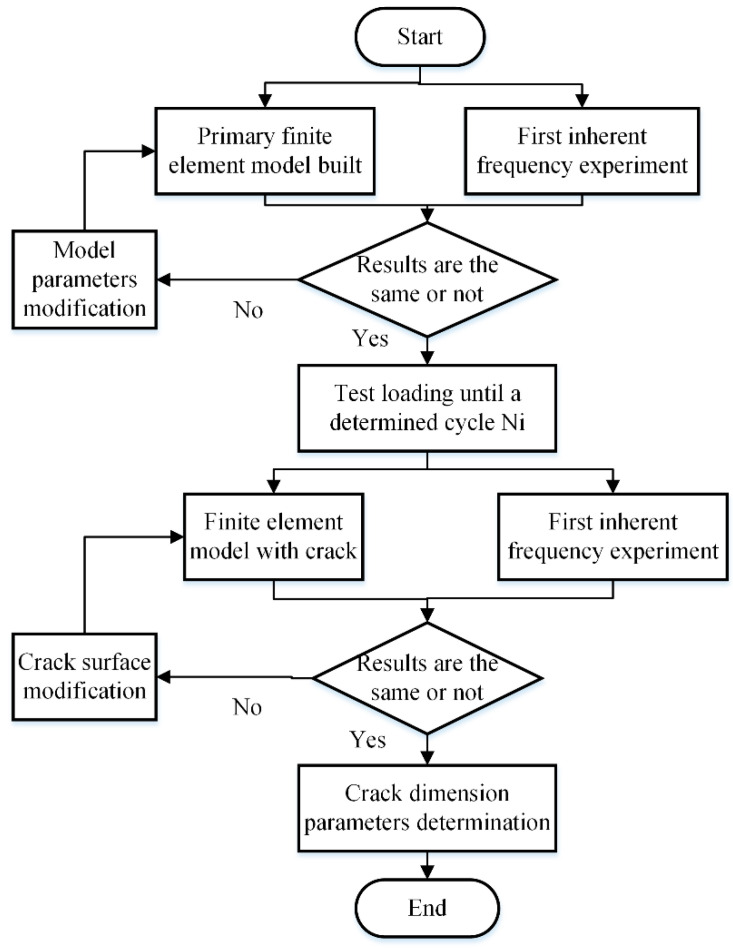
The indirect measuring method for the fatigue crack surface.

**Figure 5 materials-16-07186-f005:**
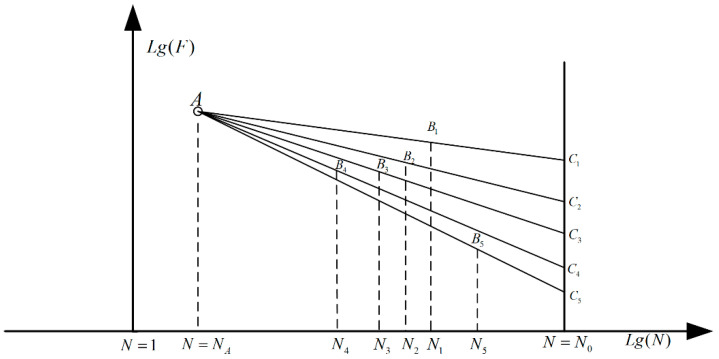
The load cycle distribution property.

**Figure 6 materials-16-07186-f006:**
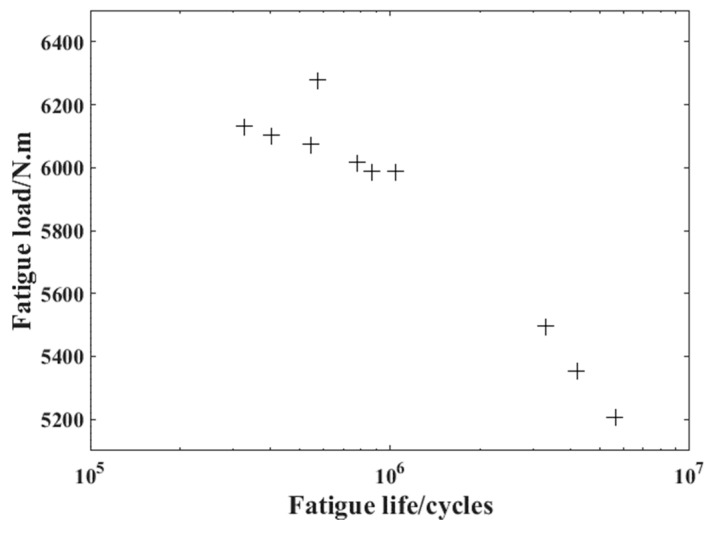
The fatigue test results of the crankshaft.

**Figure 7 materials-16-07186-f007:**
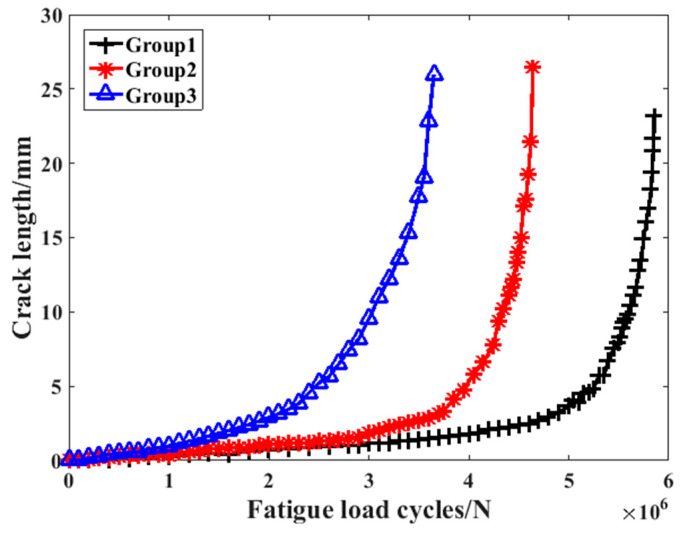
The crack–life relationship of the crankshafts in different experimental groups.

**Figure 8 materials-16-07186-f008:**
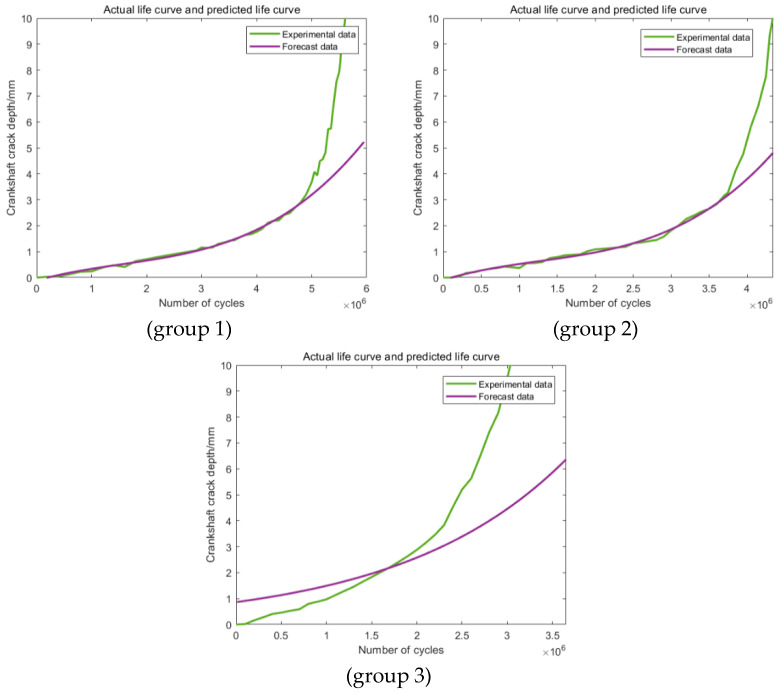
Predictions based on Range 1.

**Figure 9 materials-16-07186-f009:**
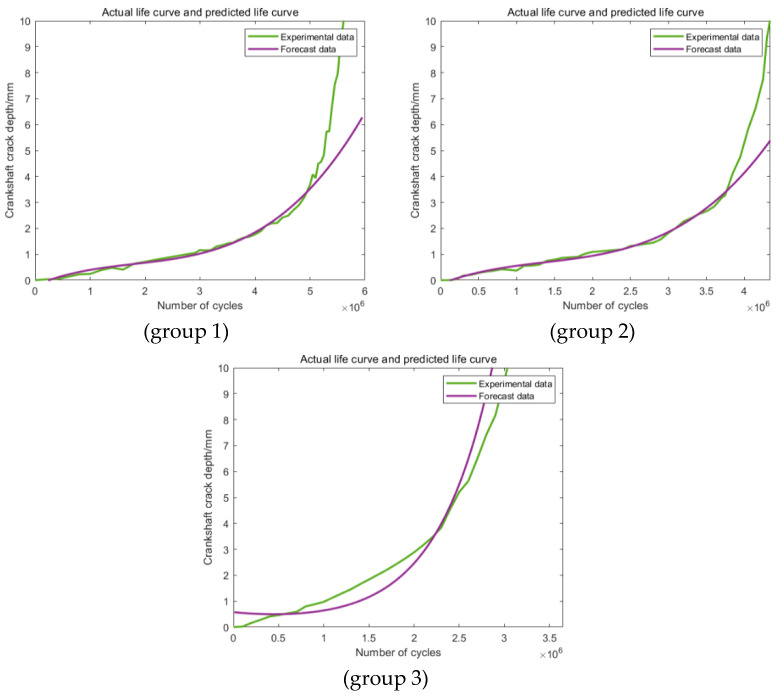
Predictions based on Range 2.

**Figure 10 materials-16-07186-f010:**
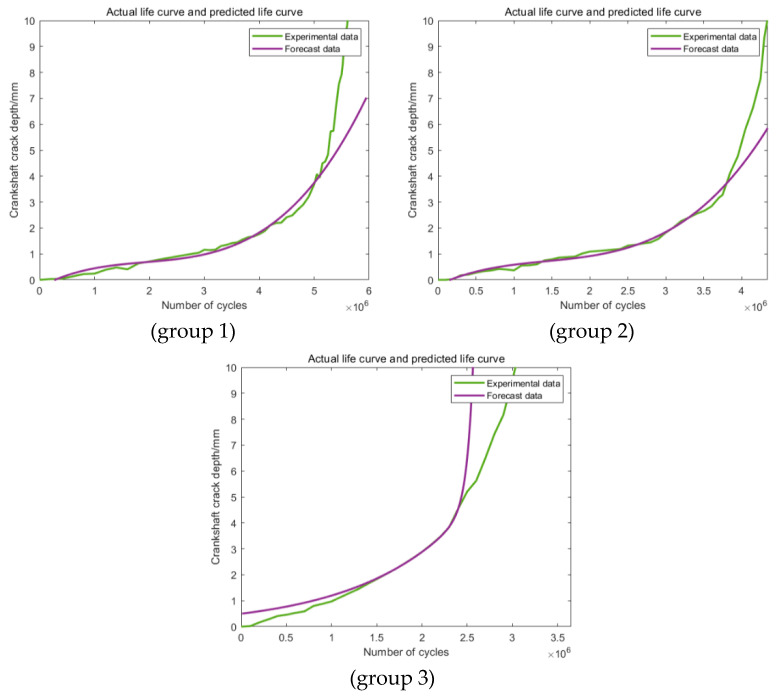
Predictions based on Range 3.

**Figure 11 materials-16-07186-f011:**
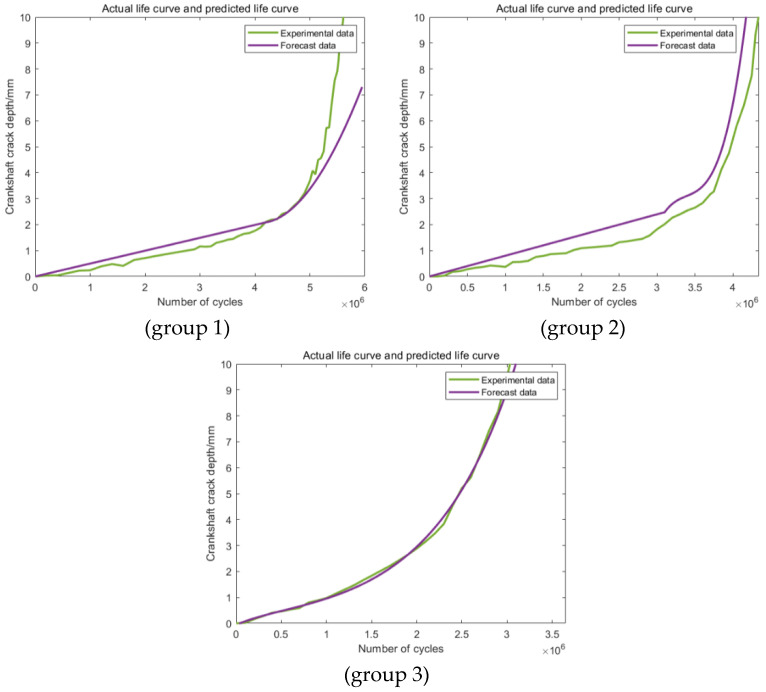
Predictions based on the modified Range 1.

**Figure 12 materials-16-07186-f012:**
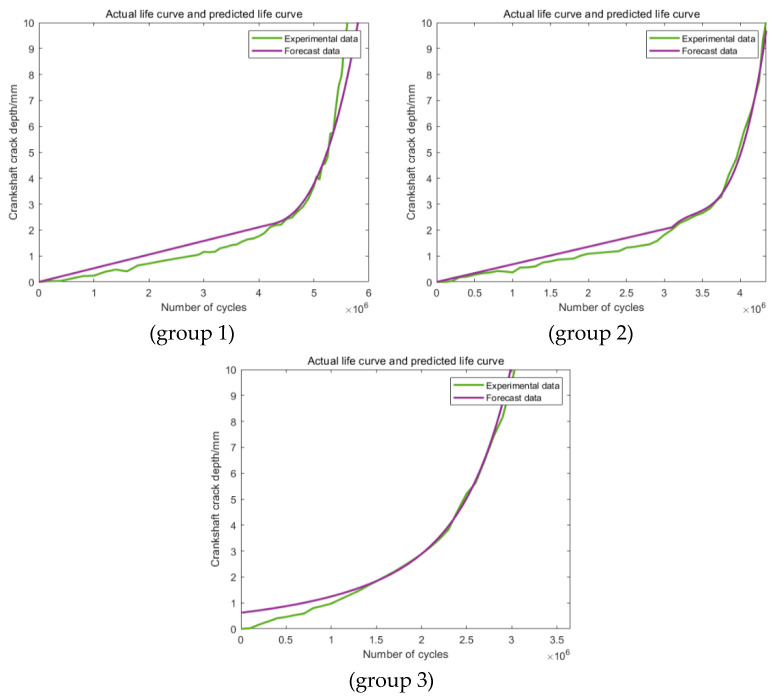
Predictions based on the modified Range 2.

**Figure 13 materials-16-07186-f013:**
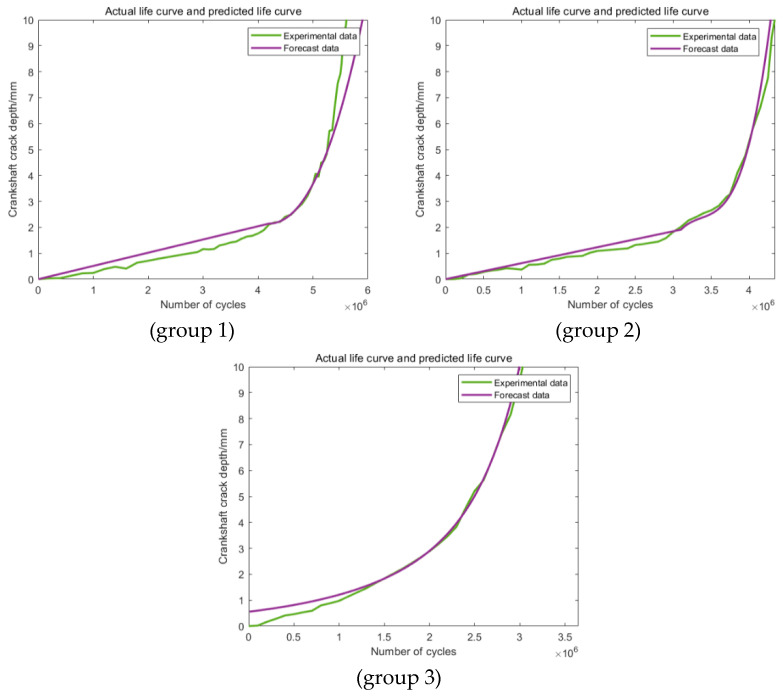
Predictions based on the modified Range 3.

**Figure 14 materials-16-07186-f014:**
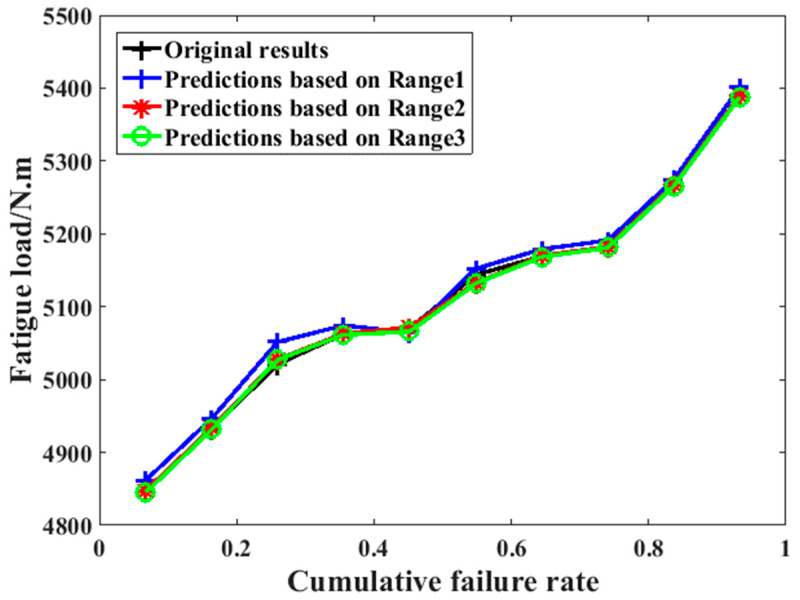
The fitting results of the fatigue limit load based on the original experimental data and predictions from different training sections.

**Table 1 materials-16-07186-t001:** The main structural parameters of the crack surface.

a/mm	b/mm
1	3.8
2	4.0
3	4.2
4	4.5
5	5.1
6	5.5
7	8.1
8	9.3
9	10.1
10	10.8

**Table 2 materials-16-07186-t002:** The influence of the crack size on the frequency.

a/mm	Frequency/Hz
0	46.2
1	46.18
2	46.14
3	46.08
4	46.04
5	45.98
6	45.86
7	45.72
8	45.56
9	45.38
10	45.19

**Table 3 materials-16-07186-t003:** The definition of the training section and prediction section.

Training Section			Prediction Section		
Range Number	Beginning	End	Range Number	Beginning	End
1	0 mm	3 mm	1	3 mm	10 mm
2	0 mm	4 mm	2	4 mm	10 mm
3	0 mm	5 mm	3	5 mm	10 mm

**Table 4 materials-16-07186-t004:** The definition of the modified training section and prediction section.

Training Section			Prediction Section		
Range Number	Beginning	End	Range Number	Beginning	End
1	1 mm	3 mm	1	3 mm	10 mm
2	1 mm	4 mm	2	4 mm	10 mm
3	1 mm	5 mm	3	5 mm	10 mm

**Table 5 materials-16-07186-t005:** The comparison of the predictions based on different models and parameters.

	Errors of Different Approaches		Timesaving Percentage	
Group	PF	UKF	Crack	Frequency
1	5.3%	9.4%	42.1%	12.9%
2	5.1%	3.1%	43.4%	6.4%
3	16.6%	1.2%	45.2%	29%
Mean value	9%	4.6%	43.6%	16.1%

## Data Availability

All data generated or analyzed during this study are included in this published article.

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
