# Peer review of "Crankshaft High-Cycle Bending Fatigue Experiment Design Method Based on Unscented Kalman Filtering and the Theory of Crack Propagation"

_materials, 2023, doi:10.3390/ma16227186_

Round 1

Reviewer 1 Report

Comments and Suggestions for Authors

1. Current similarity index shows 26% which cannot be accepted in current form. Please reduce to below 20%. Some items may cited to author's own publication works. 

2. Introduction Clarity: The manuscript's introduction needs to provide a more comprehensive overview of the importance and context of high cycle bending fatigue detection methods. Please provide some information about conventional techniques used to perform the similar failure test. 

3. The abstract is not well defined. Some information provided is not clear. Please re-structure it. 

4. Please standardize the Figure caption and table label format (Fig. x and Figure x or Tab.x and Table x), please use the standard label for both. 

5. Literature Review Enhancement: please provide more detail information about the past method used to perform the same fatigue test. 

6. In line 67, "in some cases", please provide a solid descriptions and be specific. 

7. The drawing of figure 1 does not provides clear picture. If possible please provide 3D drawing or show the actual setup.

8. Lack of clearer explanation on the Range 1, 2, and 3. Highlight the feature of results exhibit by each range.

9. In line 289, "In our published paper", this is very subjective and irresponsible. kindly provide references and citations to the materials used. 

Comments on the Quality of English Language

Required proof-reading service. 

Author Response

  1. Current similarity index shows 26% which cannot be accepted in current form. Please reduce to below 20%. Some items may cited to author's own publication works.

Answer:Thanks for your suggestion. In fact, before our submission, the whole paper (not including the reference and abstract parts) has been checked through the professional international plagiarism checker website http://hredu.qicjy.cn/gj. The repetitive rate check result for this article is 20%. The check result is uploaded as the supplyfile. If more revision is necessary, please tell us which plagiarism checker tool should be applied in checking this for further revision.

  1. Introduction Clarity: The manuscript's introduction needs to provide a more comprehensive overview of the importance and context of high cycle bending fatigue detection methods. Please provide some information about conventional techniques used to perform the similar failure test.

Answer: Thanks for your comment. In the revised manuscript, we moved the traditional experiment method description to the introduction part and added some information about this method, in this way the comprehensive description of the conventional bending fatigue test method for the crankshaft is exhibited, please check.

  1. The abstract is not well defined. Some information provided is not clear. Please re-structure it.

Answer: Thanks for your advice. We have checked and re-structure the abstract by providing more conclusions proposed in this paper, please check.

  1. Please standardize the Figure caption and table label format (Fig. x and Figure x or Tab.x and Table x), please use the standard label for both.

Answer: Thanks for remind, we have check through the whole paper and modified the labels in the text, please check.

  1. Literature Review Enhancement: please provide more detail information about the past method used to perform the same fatigue test.

Answer: Thanks for your advice. In the revised paper, we have added corresponding literatures in the introduction part, please check.

  1. In line 67, "in some cases", please provide a solid descriptions and be specific.

Answer: Thanks for your comment, we have modified this part by adding more detailed information, please check.

  1. The drawing of figure 1 does not provides clear picture. If possible please provide 3D drawing or show the actual setup.

Answer: Thanks for your advice, we have replaced this with the actual figure of the setup.

  1. Lack of clearer explanation on the Range 1, 2, and 3. Highlight the feature of results exhibit by each range.

Answer: Thanks for remind. In the revised manuscript, we have added the explanation of the ranges and the modification, as well as the feature of the prediction, please check.

  1. In line 289, "In our published paper", this is very subjective and irresponsible. kindly provide references and citations to the materials used.

Answer: Thanks for remind, we are sorry for this stupid mistake. In the revised manuscript, the references and citations has been added, please check.

Reviewer 2 Report

Comments and Suggestions for Authors

In this article, an accelerated method was proposed to shorten the time period of 

high cycle bending fatigue experiment of the crankshaft by means of UKF method. 

The fatigue limit load analysis result was proposed based on the predicted fatigue life and the modified SAFL method. The authors investigated the cycle dependence of the crankshaft crack depth precisely, and important results were provided about the durability of the crankshaft. The conclusions are consistent with the evidence and arguments presented in this article.

There are some comments:

1. The topic original and relevant in the field. However, it is a problem that there are many a figure and a description, formulas to repeat with Ref.24. Figure 1 and figure 2 are the same as figure 1 and figure 2 of Ref. 24. The formulas written from Eq.(2-1) to (2-8) seems to quoted from Ref. 24. When you quote these things, particularly figures from other articles, you must get permission of the quotation from the publisher, and must write so in the main text and/or the figure captions.

2. This study was performed and accomplished along Ref. 22-24. Therefore, in Section 1. introduction, you must explain these articles in detail to improve the understanding of the reader. 

3. Table 5 seems to fall out.

4. The differences between experimental value and calculated value gradually become small from Figure 8 to Figure 13. Describe which parameter of which formula to lowered the difference between experimental data and theoretical calculated results in detail.

5. Figure 6 : The gradient of the fatigue life dependence of the fatigue load was changed above one million cycles. Explain this reason.

6. List the references according to the MDPI style.

Comments on the Quality of English Language

The notation of Tab. # ( # : Table number) is not familiar. Change to the notation of table #.

Author Response

In this article, an accelerated method was proposed to shorten the time period of high cycle bending fatigue experiment of the crankshaft by means of UKF method. The fatigue limit load analysis result was proposed based on the predicted fatigue life and the modified SAFL method. The authors investigated the cycle dependence of the crankshaft crack depth precisely, and important results were provided about the durability of the crankshaft. The conclusions are consistent with the evidence and arguments presented in this article.

There are some comments:

  1. The topic original and relevant in the field. However, it is a problem that there are many a figure and a description, formulas to repeat with Ref.24. Figure 1 and figure 2 are the same as figure 1 and figure 2 of Ref. 24. The formulas written from Eq.(2-1) to (2-8) seems to quoted from Ref. 24. When you quote these things, particularly figures from other articles, you must get permission of the quotation from the publisher, and must write so in the main text and/or the figure captions.

Answer:Thanks for remind, in the revised manuscript, Figure 1 was replaced by the actual photo of the setup according to the other reviewer’s comment, and Figure 2 was redrawn to avoid the potential risk, please check. In addition, the repeating equations were deleted and corresponding information were added by the reference, please check.

  1. This study was performed and accomplished along Ref. 22-24. Therefore, in Section 1. introduction, you must explain these articles in detail to improve the understanding of the reader.

Answer:Thanks for your comment, in the revised manuscript, we have checked and added some more information of our previous work in the text.

  1. Table 5 seems to fall out.

Answer: Thanks for remind, we have checked and modified this.

  1. The differences between experimental value and calculated value gradually become small from Figure 8 to Figure 13. Describe which parameter of which formula to lowered the difference between experimental data and theoretical calculated results in detail.

Answer: Thanks for your comment, we have added the explanation of the modification based on the theory of fracture mechanics, as well as the crack growth speed, please check.

  1. Figure 6 : The gradient of the fatigue life dependence of the fatigue load was changed above one million cycles. Explain this reason.

Answer: Thanks for your comment, in the revised paper, we explained this phenomenon according to the fatigue crack growth speed within different stages, please check.

  1. List the references according to the MDPI style

Answer: Thanks for remind, in the revised paper, the references were modified, please check.

Round 2

Reviewer 1 Report

Comments and Suggestions for Authors

I am satisfied with the overall feedback provided by the authors. Hence, this article can be accepted for publication. However, final checking for possible grammar issue is recommended. 

Reviewer 2 Report

Comments and Suggestions for Authors

The authors revised the manuscript according to reviewer's suggestions. I accept to publish this manuscript.